# The Effect of “Proanthocyanidin” on Ischemia–Reperfusion Injury in Skeletal Muscles of Rats

**DOI:** 10.3390/medicina60050804

**Published:** 2024-05-13

**Authors:** Abdullah Özer, Başak Koçak, Şaban Cem Sezen, Mustafa Arslan, Mustafa Kavutçu

**Affiliations:** 1Department of Cardiovascular Surgery, Gazi University Medical Faculty, 06570 Ankara, Turkey; dr-abdozer@hotmail.com; 2Department of Cardiovascular Surgery, Sivas Numune Hospital, 58380 Sivas, Turkey; drbasakkocak@gmail.com; 3Department of Histology and Embriyology, Kırıkkale University, 71300 Kırıkkale, Turkey; sezenscem@gmail.com; 4Department of Anaesthesiology and Reanimation, Gazi University Medical Faculty, 06510 Ankara, Turkey; 5Department of Medical Biochemistry, Gazi University Medical Faculty, 06510 Ankara, Turkey; kavutcum@gmail.com

**Keywords:** ischemia–reperfusion, proanthocyanidin, lower extremity, rat, TBARS

## Abstract

*Background and Objectives*: Lower limb skeletal muscle ischemia–reperfusion (IR) injury is associated with increased morbidity and mortality, and it is common in several clinical situations such as aortic aneurysms repairment, peripheral arterial surgery, vascular injury repairment, and shock. Although it is generally accepted that oxidative stress mediators have a significant role in IR injury, its precise mechanism is still unknown. Anecdotally, it is sustained not only by structural and functional changes in the organ it affects but also by damage to distant organs. The purpose of this report is to illustrate the effect of proanthocyanidin on IR injury. *Materials and Methods*: In our study, 18 male Wistar albino rats were used. The subjects were divided into three groups containing six mice each (control, C; ischemia–reperfusion, IR; ischemia–reperfusion and proanthocyanidin; IR-PRO). Intraperitoneal proanthocyanidin was given to the IR and proanthocyanidin groups 30 min before laparotomy, and 1 h ischemia led to these two groups. After one hour, reperfusion started. Muscle atrophy–hypertrophy, muscle degeneration–congestion, fragmentation–hyalinization, muscle oval-central nucleus ratio, leukocyte cell infiltration, catalase enzyme activity, and TBARS were all examined in lower-limb muscle samples after one hour of reperfusion. *Results*: When skeletal muscle samples were evaluated histopathologically, it was discovered that muscle atrophy–hypertrophy, muscle degeneration–congestion, fragmentation–hyalinization, and leukocyte cell infiltration with oval-central nucleus standardization were significantly higher in the IR group than in the C and IR-P groups. Oval-central nucleus standardization was significantly higher in the IR and IR-PRO groups than in the control group. TBARS levels were significantly higher in the IR group than in the control and IR-PRO groups, while catalase enzyme activity was found to be significantly lower in the IR group than in the control and IR-PRO groups. *Conclusions*: As a consequence of our research, we discovered that proanthocyanidins administered before IR have a protective impact on skeletal muscle in rats. Further research in this area is required.

## 1. Introduction

Lower limb skeletal muscle ischemia–reperfusion (IR) injury can be seen in several clinical situations such as aortic aneurysm repairment, peripheral arterial surgery, vascular injury repairment, and shock, and it is associated with increased morbidity and mortality. Its precise mechanism is still unknown, but it is generally accepted that oxidative stress mediators have a considerable role [1].

Ischemia is the hypoperfusion of tissues induced by a limitation in arterial blood flow, and it plays a significant part in the pathogenesis of various disorders [2,3]. In addition to the cellular damage produced by ischemia alone, IR injury induced by reperfusion, which can be defined as the restoration of blood flow and reoxygenation of the ischemia-affected region [2], causes a complex process. Reperfusion subsequent ischemia is known to aggravate skeletal muscle injury.

Ischemia-induced injury can be summarized as ATP inhibition by the transition of mitochondria from aerobic to anaerobic respiration, the development of apoptosis and necrosis with calcium-dependent proteolytic enzymes activated as a result of cell depolarization with an increase in intracellular sodium and extracellular potassium, and the inhibition of other ATP-dependent mechanisms at the cellular level. While oxidative stress caused by increased reactive oxygen derivatives in the cell with reperfusion could potentially be controlled by antioxidants under normal conditions, this control ability reduces significantly during IR [2].

Ischemia–reperfusion releases reactive oxygen species (ROS) or reactive nitrogen species (RNS) like superoxide anion, hydroxyl radical, hydrogen peroxide, and peroxide nitrite. Additionally, IR induces neutrophils to cluster in a single location and become activated. Activated neutrophils contribute to ischemic injury by releasing cytotoxic free radicals and proteolytic enzymes. With high quantities of phospholipids and proteins, free radicals cause damage to cell membranes and subcellular structures, resulting in cell apoptosis and necrosis as a result of lipid peroxidation and, as a result, structural and metabolic alterations [2,4]. IR injury is characterized by structural and functional changes in the affected organ and damage to distant organs. This distant organ impact occurs as a result of free oxygen radicals released into the systemic circulation and various mediators.

Another point to consider is the oxidative stress caused by the imbalance between the production and elimination of oxidants during the release of reactive oxygen derivatives caused by IR [1,5]. Reactive oxygen derivatives cause severe damage to macromolecules due to increased oxidative stress. Membrane protein degeneration on proteins, lipid oxidation on lipids, and DNA degeneration on acids are samples of these impacts [1]. Because lipids are a crucial part of the cell membrane, the harm they are subjected to is critical [5]. Thiobarbituric Acid Reactive Substances (TBARS) assay is now employed as a marker to identify lipid oxidative damage and lipoprotein peroxidation [4,6]. It is based on the reaction of malondialdehyde (MDA) with thiobarbituric acid (TBA), which forms a pink pigment with an absorption maximum at 532 nm [7]. Catalase (CAT), which is in the enzymatic antioxidant class, has a higher antioxidant effect due to its ability to perform ROS resolution as a feature of the antioxidant class it is in [8]. The CAT enzyme, which is produced by various genes in plants such as sunflower, cotton, peas, cucumbers, rice, and pumpkins, is involved in hydrogen peroxide catalysis [9].

Proanthocyanidin is a flavonoid discovered in 1947 by French researcher Jacques Masquelier [10]. Because of their high antioxidant concentration, flavonoids, a highly essential phenol group in the human diet, have been the focus of several medical investigations. According to these research studies, proanthocyanidins, which have been shown to be a more potent antioxidant than Vitamin C and Vitamin E [11,12], are present in the structure of many plants and are mostly found in fruits such as black grapes (*Vitis vinifera*) [13] and white pine (*Pinus maritima*) [14]. It is also possible to include black oak (*Quercus marilandica*), horse chestnut (*Aesculus hippocastanum*), hawthorn (*Crataegus monogyna*), and *Vaccinium* spp. as plants in which proanthocyanidin is mainly found. In addition to its antioxidant content, its antibacterial, antiviral, anticarcinogenic, and antiallergenic effects [11,12] also exist, and it has been used in research to prevent reperfusion injury. Studies have also shown that proanthocyanidin is effective in dentin demyelination [15], catheter-related urinary tract infections [16], and some types of cancer such as hepatocellular carcinoma. Although the effect of proanthocyanidin on IR injury in different tissues and systems, such as cerebral tissue, intestinal tissue, gastric tissue, renal tissue, liver tissue, and genitourinary system, is being investigated, there are few research studies on skeletal muscle in this regard.

## 2. Materials and Method

### 2.1. Animals and Experimental Protocol

The present study was conducted at the Gazi University Animal Experiments Laboratory (Ankara, Turkey) in accordance with the ARRIVE guidelines. The study protocol was approved by the Animal Research Committee of Gazi University (G.Ü.ET-18.059). All of the animals were maintained in accordance with the recommendations of the National Institutes of Health Guidelines for the Care and Use of Laboratory Animals.

The subjects in our study were 18 Wistar Albino rats weighing between 220 and 250 g, which were nurtured under the same habitat. The subjects were kept under 20–21 °C within cycles of 12 h daylight and 12 h darkness. They were given free access to nutrition until 2 h before the anesthesia procedure and randomly separated into three equal groups of 6 animals. Ketamine anesthesia was applied prior to midline laparotomy (100 mg/kg, intraperitoneally).

**Control group (C group):** Midline laparotomy was the sole surgical procedure without any additional intervention. After 2 h of follow-up, skeletal tissue was collected. We obtained two samples from each subject for biochemical and histopathological analysis. 

**Ischemia–reperfusion group (IR group):** Midline laparotomy was performed in a similar fashion. The infrarenal aorta was left clamped for 1 h. After removing the clamp, reperfusion was established for another additional hour. At the end of 2 h, skeletal tissue was collected, and subjects were sacrificed by taking intracardiac blood. We obtained two samples from each subject for biochemical and histopathological analysis.

**Ischemia–reperfusion with proanthocyanidin group (IR-PRO group):** After following the same steps in the IR group, proanthocyanidin (Procyanidin A1, PhytoLab (Vestenbergsgreuth, Germany), CAS:103883-03-0, 5 mg) was given (10 mg/kg) intraperitoneally 30 min before the ischemia period. At the end of 2 h, skeletal tissue was collected, and subjects were sacrificed by taking intracardiac blood, as mentioned previously. We obtained two samples from each subject for biochemical and histopathological analysis.

A total of 18 different muscle tissues including control and experimental cases were processed for paraffin sections. Formalin fixation, dehydration, clearing with xylene, paraffin wax infiltration, and blocking steps were performed, respectively. Sections of four-micron thickness were taken from paraffin blocks. Then, hematoxylin and eosin stainings were performed.

The presence of muscle atrophy hypertrophy, muscle degeneration–congestion, internalization of the muscle nuclei-oval-central nucleus, leukocyte cell infiltration and fragmentation–hyalinization are evaluated according to hematoxylin–eosin sections

For hematoxylin and eosin staining, the sections were deparaffinized and hydrated. Then, the sections were stained in hematoxylin for 3 min. Afterwards, we washed the sections in tap water until the sections turned blue and were differentiated in 1% acid alcohol. The sections were washed in tap water again and then stained in eosin solution for 10 min. After washing in tap water, the sections were dehydrated and mounted.

A total of 18 skeletal muscle tissue samples separated for biochemical evaluation were first washed with cold NaCl solution (0.154 M) to discard blood contamination and then homogenized in a Diax 900; Heidolph Instruments GmbH&Co. KG, Schwabach, Germany at 1000 U for about 3 min. After centrifugation at 10,000× *g* for about 60 min, the upper clear layer was taken.

To quantify the TBARS (as malondialdehyde) levels, a thiobarbituric acid (TBA) reactive substance assay was utilized, as outlined in the literature [17]. This method involves a reaction with TBA at a temperature of 85–90 °C to determine the malondialdehyde concentration. Malondialdehyde, and similar compounds, reacts with TBA to generate a pink pigment, and its highest absorption is exhibited at 532 nm. Generally, to ensure protein precipitation, a room temperature sample is mixed with cold 20% (wt/vol) trichloroacetic acid. The precipitate is then centrifuged for 10 min at 3000 rpm and room temperature to form a pellet. A portion of the supernatant is subsequently combined with an equal volume of 0.6% (wt/vol) TBA and placed in a boiling water bath for 30 min. After cooling, the absorbance of both the sample and a blank is measured at 532 nm.

Catalase (CAT) activity was measured based on whether a decrease in absorbance was caused by the consumption of hydrogen peroxide (H_2_O_2_) at 240 nm [18]. Furthermore, GST activity was determined based on the measurement of absorbance increase at 340 nm due to a reduction in dinitrophenyl glutathione (DNPG), as described by Habig et al. [19]. Activity measurements were performed using the ε value of the DNPG complex.

The protein amounts of the samples were determined using the Lowry method with bovine serum albumin used as the standard protein [20]. The results were expressed as IU/mg protein for enzymes and nmo/mg protein for TBARS.

### 2.2. Statistical Analysis

The Statistical Package for the Social Sciences (SPSS, Chicago, IL, USA) 20.0 for Windows was used. The Kolmogorov–Smirnov test was used for analyzing each category variable. Biochemical and histopathological parameters were tested by using the Kruskal–Wallis test, Bonferroni correction test and Mann–Whitney U-test. A statistical value of less than 0.05 was considered significant. All values were expressed as mean ± standard error (Mean ± SE).

## 3. Results

### 3.1. Histopathological Results

The histological parameters muscle atrophy–hypertrophy, muscle degeneration–congestion, the internalization of the muscle nuclei oval-central nucleus, leukocyte cell infiltration, and fragmentation–hyalinization varied considerably across groups (*p* = 0.003, *p* = 0.002, *p* = 0.001, *p* = 0.006, *p* = 0.004, respectively), (Table 1A,B, Figure 1 and Figure 2). Muscle atrophy–hypertrophy was significantly higher in the IR group than in the C and IR-PRO groups (*p* = 0.001, *p* = 0.011, respectively). The muscle degeneration–congestion of the IR group was substantially greater than in the C and IR-PRO groups (*p* = 0.001, *p* = 0.005, respectively). In the IR group, the internalization of the muscle nuclei oval-central nucleus was notably higher than in the C and IR-PRO groups (*p* < 0.0001, *p* = 0.024, respectively). Additionally, the internalization of the muscle nuclei oval-central nucleus was extensively elevated in the IR-PRO group compared with the C group (*p* = 0.024). Leukocyte cell infiltration and fragmentation–hyalinization were also considerably higher in the IR group than in the C and IR-PRO groups (*p* = 0.002, *p* = 0.023, respectively), (*p* = 0.002, *p* = 0.006, respectively) (Table 1A,B, Figure 1 and Figure 2).

### 3.2. Biochemical Results

When the levels of TBARS in muscle tissue were measured within the groups, there was a substantial variance. The IR group’s TBARS level was found to be considerably higher than the control group’s (*p* < 0.0001). The TBARS level was found to be significantly lower in the IR-PRO group compared to the IR group (*p* < 0.0001; Table 2).

In terms of CAT enzyme activity in muscle tissue, there was a remarkable difference between the groups (*p* < 0.0001). The IR group’s CAT enzyme activity was found to be considerably decreased compared with the control group’s (*p* < 0.0001). CAT enzyme activity was notably increased in the IR-PRO group compared with the IR group (*p* < 0.0001), although it was similar in the C and IR-PRO groups (*p* = 0.284; Table 2).

## 4. Discussion

Due to nonignorable clinical outcomes of IR injury, various substances, including antioxidants, statins, and anesthetics, have been investigated and suggested for the preventative therapeutic use of IR injury in studies [21]. According to our knowledge, there are few investigations of proanthocyanidins’ effect on lower extremity skeletal muscle IR injury. Our study hypothesized that proanthocyanidin would have a protective effect on IR injury in the skeletal muscle of the lower extremities. Our conclusions support our theoretical basis.

As mentioned before, IR injury increases ROS and RNS and decreases antioxidant enzyme levels and expression, according to several studies [22]. Numerous enzymes perform as intracellular antioxidants to protect cells from IR-induced oxidative damage [23]. In this manuscript, we determined the CAT enzyme activity for cellular antioxidant defense functions. In this regard, our study supported the literature by increasing the CAT enzyme in the IR-PRO and C groups compared to the IR group. Güler et al. investigated proanthocyanidin and its protective effect on myocardial ischemia–reperfusion [13] and emphasized increased CAT enzyme activities in the control and proanthocyanidin groups. In addition, despite the variety of studies on CAT enzyme and ischemia–reperfusion injury [24,25], there are limited research studies on its relationship with proanthocyanidin according to our knowledge [26].

Based on our observations, another affirming finding is the difference in TBARS level for determined lipid peroxidation—a process by which free radicals destroy lipids’ carbon–carbon double bonds [7], which were elevated in the IR group and reduced in the C and IR-PRO groups. Our TBARS results align with those in the literature [27,28].

Histological evaluation was in line with biochemical findings. Intraperitoneal proanthocyanidin treatment before ischemia in rats reversed IR injury in skeletal muscle histology. Our data verified proanthocyanidin’s protection. According to histological inspection, IR damage was associated with muscle degeneration–congestion, fragmentation–hyalinization, leukocyte cell infiltration, muscle atrophy–hypertrophy, and internalization of oval-central nuclei, and proanthocyanidin administration prior to ischemic injury inhibited these changes and preserved skeletal muscle.

These changes demonstrated that proanthocyanidin can minimize the IR-induced destruction of cells, protecting muscles from IR-induced damage.

We preferred 10 mg/kg, i.p. proanthocyanidin type A1 for our research. Despite the previously referenced studies regarding different amounts of proanthocyanidin applied in different clinical situations, to our knowledge, no investigation has been conducted to compare the efficacy of proanthocyanidin types in lower limb IR injury. In addition, it might be beneficial to investigate whether the application methods (i.p vs. p.o) and duration of proanthocyanidin could change its effect on the same clinical situations such as IR injury.

Following extensive randomized clinical trials completed on human subjects, the findings must be applied to practical medical practice.

## 5. Conclusions

In summary, we believe our research showed that intraperitoneal proanthocyanidin has a protective effect on IR injury of skeletal muscle with a possible mechanism by reducing lipid peroxidation and increasing antioxidant enzyme activity. Examining the literature reveals that there are numerous research studies investigating proanthocyanidin. The existence of several types of proanthocyanidin, as well as changes in the duration of administration, dosage, and form, restrict our ability to provide a precise analysis. Future research is required to determine the IR injury effect and proanthocyanidin efficacy.

## Figures and Tables

**Figure 1 medicina-60-00804-f001:**
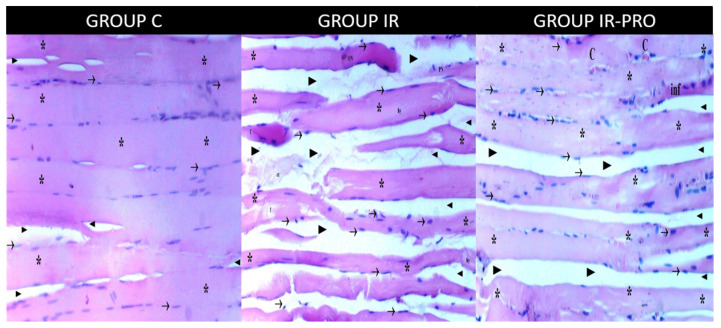
Skeletal muscle longitudinal section light microscopy for the C, IR and IR-PRO groups. (H&E: hematoxylin and eosin X100). →: peripheric nucleus; *: muscle fibers; ►: intercellular space; H: hypertrophy; NF: necrotic fibrillar area; C: central nucleus; ON: oval nucleus; f: fragmentation; hy: hyalinization; p: picnotic nucleus; O: edema; inf: infiltration; ct: cutaneus tissue. The muscle fiber, a poly-nucleated, syncytial-like structure, resembles an elongated, slender tube. Its sarcoplasm, with a pale pink shade, houses elongated and thin sarcolemmal nuclei. These nuclei are strategically positioned, aligning with the length of the fiber, adding to its unique structure. In the IR group, increased intercellular space, fragmentation–hyalinization, and picnotic and oval nuclei are visible. The IR-PRO group has a similar light microscopic view with the C group due to the protective effect of proanthocyanidin.

**Figure 2 medicina-60-00804-f002:**
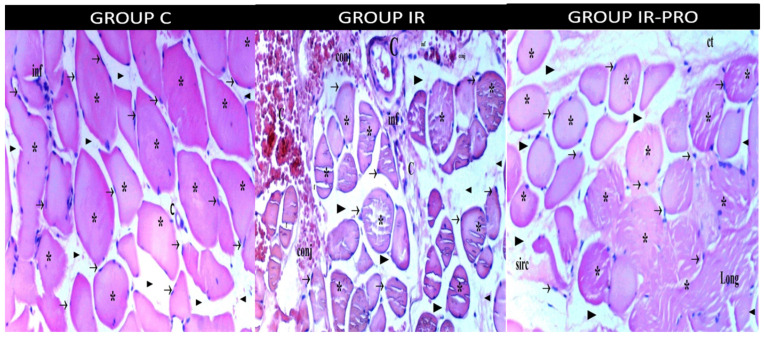
Skeletal muscle cross-section light microscopy for the C, IR and IR-PRO groups. (H&E: hematoxylin and eosin X100). →: peripheric nucleus; *: muscle fibers; ►: intercellular space; H: hypertrophy; NF: necrotic fibrillar area; C: central nucleus; ON: oval nucleus; f: fragmentation; hy: hyalinization; p: picnotic nucleus; O: edema; inf: infiltration; ct: cutaneus tissue; conj: conjestion; sirc: sircular; long: longitudinal. When viewed in a cross-section, the norm for adult myocytes is not entirely round but instead polygonal, resulting in a complex profile. The nuclei, often positioned near the sarcolemma, number four to six per cell when cut in a transverse slice. In this view, congestion and central nuclei are visible. Notably, the findings in the longitudinal section are mirrored in the cross-section, with the IR-PRO group showing slight microscopic changes compared to the IR group, which was potentially due to the proanthocyanidin effect.

**Table 1 medicina-60-00804-t001:** A: Rat extremity muscle tissue histopathologic data [mean ± SE]. B: Rat extremity muscle tissue histopathologic 4-point scoring data.

**A**																		
							**C** **Group** **(n = 6)**			**IR** **Group** **(n = 6)**			**IR-PRO** **Group** **(n = 6)**				***p*** ******	
**Muscle atrophy–hypertrophy**							0.17 ± 0.17 *			1.33 ± 0.21			0.50 ± 0.22 *				0.003	
**Muscle degeneration–congestion**							0.33 ± 0.21 *			1.83 ± 0.31			0.67 ± 0.21 *				0.002	
**Internalization of muscle nuclei oval-central nucleus**							0.17 ± 0.17 *			1.50 ± 0.22			0.83 ± 0.17 *, &				0.001	
**Fragmentation–hyalinization**							0.33 ± 0.21 *			1.83 ± 0.31			0.83 ± 0.31 *				0.006	
**Leukocyte cell infiltration**							0.33 ± 0.21 *			1.50 ± 0.22			0.50 ± 0.22 *				0.004	
**B**																		
	**C** **Group** **(n = 6)** **(I, II, III, IV, V, VI)**						**IR** **Group** **(n = 6)** **(I, II, III, IV, V, VI)**						**IR-PRO** **Group** **(n = 6)** **(I, II, III, IV, V, VI)**					
	I	II	III	IV	V	VI	I	II	III	IV	V	VI	I	II	III	IV	V	VI
**Muscle atrophy–hypertrophy**	1	0	0	0	0	0	2	1	1	2	1	1	1	1	1	0	0	0
**Muscle degeneration–congestion**	1	1	0	0	0	0	3	1	2	2	1	2	1	1	1	1	0	0
**Internalization of muscle nuclei** **Oval-central nucleus**	1	0	0	0	0	0	1	2	1	2	2	1	1	1	1	1	1	0
**Fragmentation–hyalinization**	1	1	0	0	0	0	2	1	2	2	1	3	1	1	1	2	0	0
**Leukocyte cell infiltration**	1	1	0	0	0	0	2	1	2	2	1	1	0	1	1	0	0	1

*p* ** Significance level with Kruskal–Wallis test *p* < 0.05. * *p* < 0.05: Compare to IR group. & *p* < 0.05: Compare to C group.

**Table 2 medicina-60-00804-t002:** Rat extremity muscle tissue TBARS and CAT enzyme activity data [mean ± SE].

	C Group (n = 6)	IR Group (n = 6)	IR-PRO Group (n = 6)	*p* **
**TBARS (nmol/mg pr)**	26.58 ± 5.58 *	62.27 ± 3.76	34.00 ± 4.73 *	<0.0001
**CAT (IU/mg pr)**	8.73 ± 1.09 *	5.16 ± 0.51	8.21 ± 0.67 *	<0.0001

*p* ** Significance level with Kruskal–Wallis test *p*< 0.05. * *p* < 0.05: Compare to IR group.

## Data Availability

The datasets used and/or analyzed during the current study are available from the corresponding author on reasonable request.

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
