# Peer review of "The Effect of “Proanthocyanidin” on Ischemia–Reperfusion Injury in Skeletal Muscles of Rats"

_medicina, 2024, doi:10.3390/medicina60050804_

Round 1

Reviewer 1 Report

Comments and Suggestions for Authors

I have reviewed in detail the paper entitled: “The effect of “Proanthocyanidin” on ischemia-reperfusion injury in skeletal muscles of rats”. In this article, the authors evaluated the effects of Proanthocyanidin administration in an ischemia-reperfusion injury model. I comment the following: 

1. It is not clear to me what the main objective of the work is. The authors only mention that the number of research on skeletal muscle in this regard is few, and that is not a valid objective.

2. Why use a rat model? Is it the ideal animal model for your work? What previous works were you based on to calculate the ischemia time, dose and route of administration of Proanthocyanidin that can support your model?

3. How did you sacrifice the animals?

4. How did you evaluate the histological parameters: muscle atrophy-hypertrophy, muscle degeneration-congestion, the internalization of muscle nuclei oval-central nucleus, and fragmentation-hyalinization? You don't describe the process, what did you measure? The thickness, the increase in the number of muscle fibers, the space between them?

5. In Table 1, you only put the mean standard deviation, but you did not mention the units (cm, mm, cells, diameter, etc.)

6. There are too many figures; I consider that they could be reduced to two figures if they are grouped in Fig. 1 longitudinal section (groups C, IR, and IR-PRO) and Fig. 2 cross-section (groups C, IR, and IR-PRO). It would be easier for the reader to be able to appreciate the histological changes in each group.

7. The discussion must be improved, it is very poor, they did not compare their results with those obtained in other previous works, and they also did not mention the strengths and limitations of their work.

Author Response

Dear editor and reviewers;

We thank the reviewer for their positive and constructive comments,and also we thank editor for this decision.

You may find our point-to-point responses to the comments/suggestions by the reviewers below.We also highlighted the changes in yellow within the text.

Best regards,

Mustafa Arslan, M.D.

Reviewer 2 Report

Comments and Suggestions for Authors Özer, A, et al., have examined the effect of proanthocyanidin on ischemia-reperfusion injury in the skeletal muscles of rats. The study aims to investigate the protective impact of proanthocyanidin on skeletal muscle in rats subjected to ischemia-reperfusion injury. The authors conducted experiments on male Wistar albino rats and divided them into three groups: Group Control-C, Group Ischemia-Reperfusion (IR), and Group Ischemia-Reperfusion and Proanthocyanidin (IR-PRO). Proanthocyanidin was administered intraperitoneally to the IR and IR-PRO groups before inducing ischemia. After one hour of reperfusion, various parameters such as muscle atrophy-hypertrophy, muscle degeneration-congestion, fragmentation-hyalinization, muscle oval-central nucleus ratio, leukocyte cell infiltration, catalase enzyme activity, and TBARS (Thiobarbituric Acid Reactive Substances) levels in the lower-limb muscle samples were examined. The results showed that proanthocyanidins administered before IR had a protective impact on skeletal muscle in rats. The study suggests that further research is needed in this area. This is a commendable piece of work, and the study was conducted meticulously. Here are my comments for the authors:
  1. In the material methods (animal study protocol), in paragraph 7 after the word "infiltration," there is "ve." Could you please clarify what "ve" stands for?
  2. The legends for the H & E images in Figure 1 are missing. I would kindly request an explanation for all H & E images to better understand the results.
  3. In this paper, there seems to be a lack of consistency in the formatting. Some figure legends are written in uppercase, while others are written in lowercase. It would be beneficial to maintain consistency throughout the article.
  4. I would like to suggest that the authors rewrite the discussion and conclusion sections based on their recommendations for further elucidation and future experiments. This would provide clarity on what is required to further explore this area of research.

Thankyou so much.

Regards

Author Response

(The authors gave the same response as above.)

Round 2

Reviewer 1 Report

Comments and Suggestions for Authors

The authors resolved the questions raised and the article improved